# Recipes for improper ferroelectricity in molecular perovskites

Hanna L.B. Boström[1], Mark S. Senn [1,2] & Andrew L. Goodwin [1]

The central goal of crystal engineering is to control material function via rational design of structure. A particularly successful realisation of this paradigm is hybrid improper ferroelectricity in layered perovskite materials, where layering and cooperative octahedral tilts combine to break inversion symmetry. However, in the parent family of inorganic $ABX_3$ perovskites, symmetry prevents hybrid coupling to polar distortions. Here, we use group-theoretical analysis to uncover a profound enhancement of the number of improper ferroelectric coupling schemes available to molecular perovskites. This enhancement arises because molecular substitution diversifies the range of distortions possible. Not only do our insights rationalise the emergence of polarisation in previously studied materials, but we identify the fundamental importance of molecular degrees of freedom that are straightforwardly controlled from a synthetic viewpoint. We envisage that the crystal design principles we develop here will enable targeted synthesis of a large family of new acentric functional materials.

[1] Department of Chemistry, University of Oxford, Inorganic Chemistry Laboratory, South Parks Road, OX1 3QR Oxford, UK. [2] Department of Chemistry, University of Warwick, Gibbet Hill, CV4 7AL Coventry, UK. Correspondence and requests for materials should be addressed to M.S.S. (email: m.senn@warwick.ac.uk) or to A.L.G. (email: andrew.goodwin@chem.ox.ac.uk)

**F**erroelectricity, i.e., the presence of a switchable electric polarisation, is an important property with many technological applications[1]. An early canonical ferroelectric was $BaTiO_3$, with the broader family of $ABX_3$ perovskite oxides now known to include a variety of ferroelectric systems[2–5]. The ferroelectric response of $BaTiO_3$ originates from a second-order Jahn–Teller (SOJT) effect, in which the B-site cation $Ti^{4+}$ displaces from the centre of its $TiO_6$ coordination environment[2,6]. As this polar instability also acts as the primary order parameter, $BaTiO_3$ is a proper ferroelectric. Despite the continuing discussion regarding the origin of polar coupling in $BaTiO_3$[7], the general SOJT mechanism at play is rare and its generalisation is challenging, as it requires a $d^0$ electronic configuration[6]. Hence, design approaches for new ferroelectric materials based on SOJT instabilities are limited and, moreover, the mechanism is difficult to couple with spin-active $d$-electron configurations, which has largely prevented its exploitation in the search for magnetoelectric multiferroics[8]. In fact, ferroelectricity is comparatively rare in bulk perovskite materials and $BaTiO_3$ is much more of an exception than a rule[9].

It is in this context that the concept of hybrid improper ferroelectricity is especially appealing[10]. A necessary condition for ferroelectricity is the presence of a polar space group, which in turn requires broken inversion symmetry. Simple inorganic perovskites contain two crystallographically distinct inversion centres (at the A- and B-site, respectively) and in $BaTiO_3$, the zone-centre polar mode breaks both of these, driving the polarity[11]. Alternatively, in favourable cases, a combination of two or more modes—each non-polar in their own right—may collectively lift inversion symmetry and give rise to a polar secondary order parameter[10,12]. This so-called hybrid improper ferroelectricity mechanism is attractive from a crystal engineering perspective because it lends itself to design rules via group-theoretical analysis[10]. In addition, the mechanism does not preclude magnetic order[10,13]. Hence, using group-theoretical methods, it is possible to enumerate the symmetry breaking caused by given distortions of an aristotype and thereby predict the propensity for the formation of acentric structures. For simple inorganic perovskites, the accessible degrees of freedom—cooperative first-order Jahn–Teller (FOJT) distortions and octahedral tilting—all preserve the inversion centre of the B-site, and in order to enable hybrid improper ferroelectricity, additional symmetry breaking in the form of A-site cation order or layering is needed[13–15].

A recent development in the broader field is the increased interest in molecular perovskite analogues[16]. These are solid compounds with the same $ABX_3$ stoichiometry of conventional perovskites, but where A or X (or both) are molecular ions. This populous class of materials may be categorised according to the nature of the anionic linker: topical families include the organic–halide perovskites[17,18], metal formates[19,20], Prussian blue analogues[21,22], azides[23,24], dicyanamides[25,26], hypophosphites[27], thiocyanates[28,29], and dicyanometallates[30,31]. The enhanced structural flexibility allowed by molecular species enables additional degrees of freedom unfeasible in conventional inorganic perovskites: unconventional tilts[30,32], columnar shifts[33], and multipolar order (Fig. 1)[34,35]. The first two of these correspond to rigid-unit modes (RUMs)[36]—i.e., phonon modes which propagate without deforming $BX_6$ coordination geometries[37,38]. The larger number of RUMs in molecular perovskites is conceptually related to the additional flexibility driven by reduced connectivity in Ruddlesden–Popper phases[39]. By contrast, the presence of multipolar degrees of freedom reflects reorientations of non-spherical molecular A-site cations[34,35,40].

In this paper, we show how these new types of structural degrees of freedom can combine to break inversion symmetry, hence establishing a new set of design rules for engineering acentric molecular perovskites. First, we introduce our general recipe for designing acentric materials. Second, we classify the various symmetry breaking ingredients accessible to molecular perovskites in terms of the irreducible representations of the high-symmetry perovskite aristotype. Third, we use these ingredients to construct the key ferroelectric coupling schemes. And, fourth, we illustrate how this approach can be used to rationalise the emergence of polarisation in a number of previously reported systems.

## Results

**Polarisation from trilinear coupling**. The fundamental idea of our approach is to identify two distortions ($A$ and $B$) of the parent $Pm\bar{3}m$ aristotype that are inherently non-polar but which, when combined, give rise to an additional polar degree of freedom ($P$) in the hettotype (child structure). In a Landau-style expansion of the free energy about the parent structure, this produces a third-order (trilinear) term $\beta ABP$. As $A$ and $B$ are inherently non-zero if they are unstable with respect to the parent phase then—irrespective of the sign of the coefficient $\beta$—$P$ will also adopt a non-zero (positive or negative) value in order to stabilise the free energy. We can use the idea of invariants analysis[41] to identify the permissible third-order terms in the general free energy expansion; i.e., identify for which combinations of $A$ and $B$ one would expect coupling to $P$. For the zone-boundary/zone-centre distortions that we consider here, we find that such couplings may in fact be identified by inspection, as detailed in the following paragraphs.

Each distortion we consider can be described as transforming as an irreducible representation (irrep) of the parent space group $Pm\bar{3}m$ (Table 1, see Supplementary Fig. 1 and Supplementary Table 1 for a more comprehensive list). The irrep labels are of the form $\mathbf{k}_{\#}^{\pm}$, where the symbol $\mathbf{k}$ denotes the propagation or wave vector of the distortion with respect to the parent structure. In our study we consider $\mathbf{k} = [0, 0, 0]$ ($\Gamma$), $\left[\frac{1}{2}, \frac{1}{2}, \frac{1}{2}\right]$ (R), $\left[\frac{1}{2}, \frac{1}{2}, 0\right]$ (M), $\left[0, \frac{1}{2}, 0\right]$ (X), and symmetry equivalents thereof—these being the distortion periodicities most relevant to real examples. The + or − sign generally denotes that either one or the other inversion centre, related to each other by a origin shift (i.e., a translation by $\frac{1}{2}, \frac{1}{2}, \frac{1}{2}$), is broken. Clearly, any collective ordering or distortion that fails to break both these inversion centres at once cannot lead to an improper ferroelectric coupling, and it is this point that underlies our arguments based on invariants analysis presented below.

As any term that features in the free energy expansion about the parent phase must conserve both crystal momentum and parity with respect to the inversion symmetry, it can quickly be seen that any two distortions transforming as $X^+$ and $X^-$, or as $M^+$ and $M^-$, or as $R^+$ and $R^-$ must couple to a non-centrosymmetric (and in general polar) distortion $P$, which itself transforms as $\Gamma^-$. On the other hand, any two distortions to the parent phase that have either a different propagation vector and/or the same sign with respect to inversion parity cannot produce such a coupling to $P$. Hence, combinations such as $X^+$ and $R^-$, or $M^-$ and $M^-$ etc., may be immediately discounted. As a final point, we note that higher-order coupling terms such as ($X^+$, $M^+$, $R^-$), ($X^-$, $M^-$, $R^-$), and so on also produce a fourth-order coupling in $P$, provided that the rules for preservation of crystal momentum and parity are observed.

**Distortions in molecular perovskites**. In the context of molecular perovskites, the relevant collective distortions involve conventional tilts, unconventional tilts, columnar shifts, FOJT effects, and multipolar A-site order. The combinations of distortions capable of driving polarity are shown in Fig. 2, with an example of

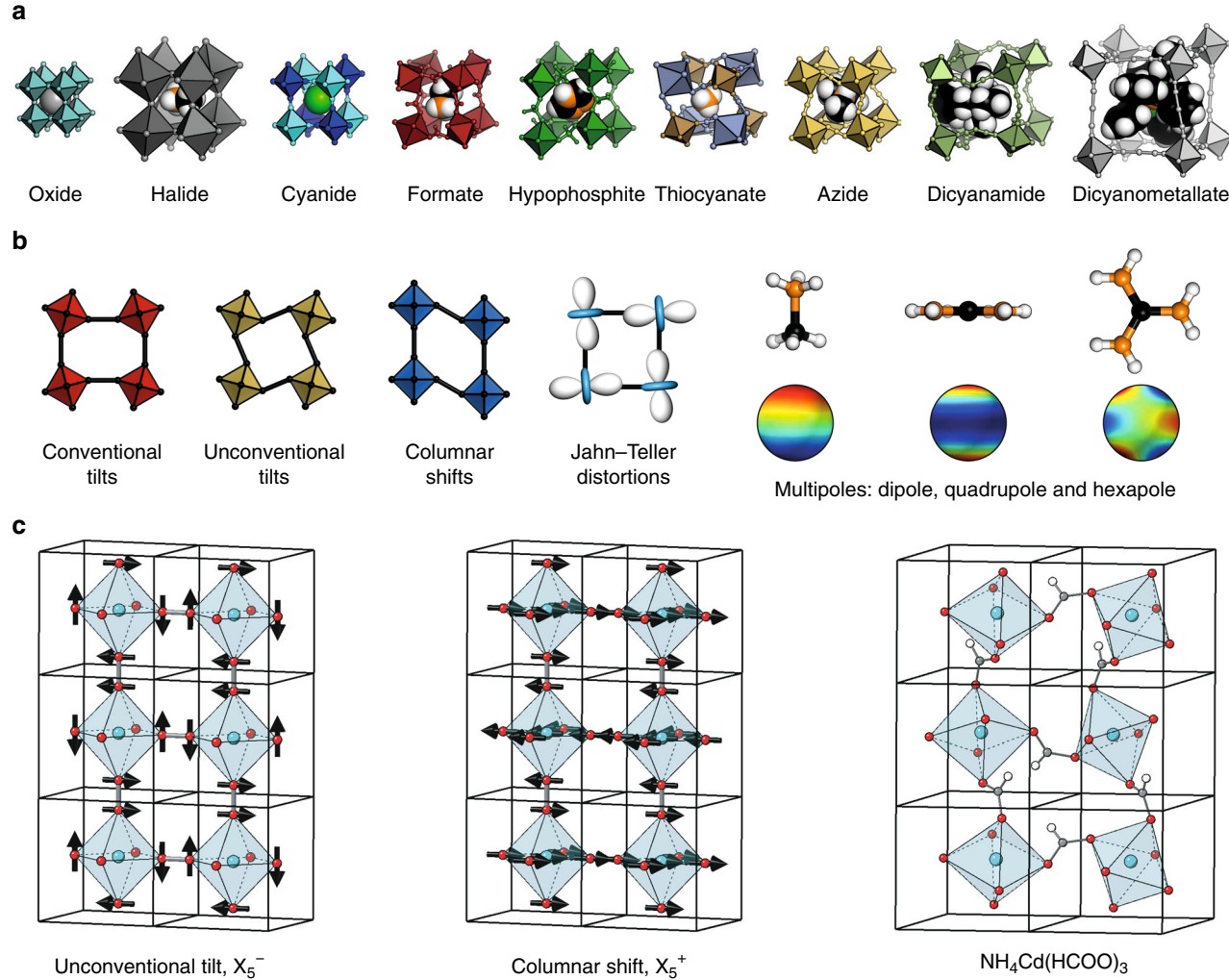

**Fig. 1** Molecular perovskites and their degrees of freedom. **a** A perovskite oxide with the molecular congeners shown to scale. In all cases, the A-site cation is shown as spacefilling, with carbon in black, hydrogen in white, nitrogen in orange, and phosphorus in green. **b** Schematic illustrations of the various degrees of freedom accessible to molecular perovskites. **c** A combination of two such distortions—an unconventional tilt (left) and columnar shift (centre) —is responsible for the crystal symmetry of $[NH_4]Cd(HCOO)_3$ (right). Note that the unconventional tilt results in some neighbouring coordination octahedra rotating in the same sense, which is possible only because the X-site anions are molecular

**Table 1 The irreps corresponding to the different distortions considered**

| Distortion | Irreps |
|---|---|
| Conventional tilting | $M_2^+$, $R_5^-$ |
| Unconventional tilting | $\Gamma_4^+$, $X_{1,5}$, $M_5^+$ |
| Columnar shifts | $\Gamma_{3,4,5}^+$, $X_5^+$, $M_2^+$ |
| Jahn–Teller distortions | $M_3^+$, $R_3^+$ |
| Quadrupolar A-site order | $\Gamma_{3,5}^+$, $X_{2,5}^+$, $M_{1,2,4,5}^+$, $R_5^+$ |
| Dipolar A-site order | $\Gamma_4^-$, $X_{3,5}^-$, $M_{3,5}^-$, $R_4^-$ |

a representative child space group given for each specific combination. For analysis of related third- and fourth-order coupling schemes, see Supplementary Tables 2 and 3. Our main result is the clear distinction in number of possibilities for inversion symmetry breaking in molecular perovskites relative to their conventional ceramic counterparts (inset in Fig. 2). Indeed, we find that either multipolar order (driven by molecular substitution on the A-site) or the activation of columnar shifts (driven by molecular substitution on the X-site) is by itself sufficient to break inversion symmetry when correctly coupled to any other order

parameter. Hence, in the design of ferroelectric molecular perovskites, molecular substitution need involve only one site (A or X) and polar ground states are theoretically possible both for A-site-only substituted perovskites (e.g., the organic–halide perovskites) and for their X-site-only substituted cousins (e.g., Prussian blue analogues).

Arguably one of the easiest distortions to control from a materials design perspective is that of A-site multipolar order. Many common organic cations are polar—their charge being localised on a single atom. Moreover, both rod-like (prolate) and disc-like (oblate) cations possess quadrupole moments. This includes common cations such as methylammonium ($CH_3NH_3^+$), hydrazinium ($NH_2NH_3^+$), guanidinium ($C(NH_2)_3^+$), and imidazolium. Higher-order multipoles are also possible, such as the hexapole moment of guanidinium and the octupole moment of ammonium and tetramethylammonium. Table 1 makes clear that equivalent types of zone-boundary dipolar and quadrupolar order transform as irreps with opposite sign with respect to inversion parity. Hence, the incorporation of cations which support both dipole and quadrupole moments gives access to a greatly increased number of possible improper couplings to bulk polarisation.

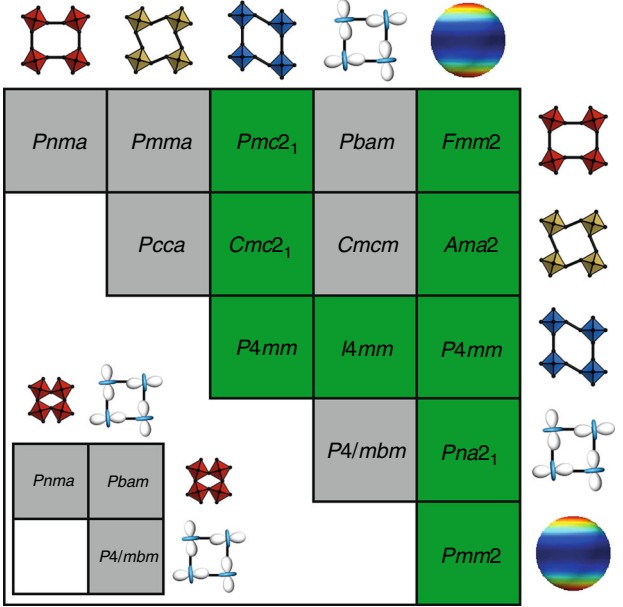

**Fig. 2** Coupling schemes in molecular perovskites. The accessible distortion types are given at the top of each column and the right of each row: conventional tilts, unconventional tilts, columnar shifts, Jahn–Teller distortions, and multipole ordering. For each combination of distortions, a representative space group is shown and the colour indicates whether coupling of the two distortions can ever drive a polar distortion (green) or not (grey). The inset shows the corresponding coupling scheme for conventional inorganic perovskites

**Table 2** A summary of known polar molecular perovskites, their crystal symmetries, and the corresponding distortion mode irreps

| Compound | Space group | Irreps | | | Ref. |
|---|---|---|---|---|---|
| $NH_4Cd(HCOO)_3$ | $Pna2_1$ | $X_5^-$ | $X_5^+$ | | 42 |
| $[C(NH_2)_3]Cu(HCOO)_3$ | $Pna2_1$ | $M_3^+$ | $X_5^+$ | $R_5^-$ | 44 |
| $[C(NH_2)_3]Cr(HCOO)_3$ | $Pna2_1$ | $M_3^+$ | $X_5^+$ | $R_5^-$ | 50 |
| $[NH_3NH_2]Mn(HCOO)_3$ | $Pna2_1$ | $M_2^+$ | $X_5^+$ | $R_5^-$ | 56 |
| $[NH_3NH_2]Zn(HCOO)_3$ | $Pna2_1$ | $M_2^+$ | $X_5^+$ | $R_5^-$ | 56 |
| $[EtNH_3]Mn(HCOO)_3$ | $Pna2_1$ | $M_2^+$ | $X_5^+$ | $R_5^-$ | 45 |
| $[PrPEt_3]Mn(dca)_3$ | $P2_12_12_1$ | $M_2^-$ | $X_5^-$ | $R_5^-$ | 54 |
| $[MeOCH_2PEt_3]Mn(dca)_3$ | $P2_12_12_1$ | $M_2^-$ | $X_5^-$ | $R_5^-$ | 54 |

of the same family, which immediately identifies the importance of a collective Jahn–Teller distortion $(M_3^+)$ in the coupling scheme. The multipolar order associated with the particular alignment of the Gua cation couches in fact two order parameters: the quadrupolar ordering $X_5^+$ (orientation of the guanidinium plane normal) and the hexapolar ordering $R_5^-$ (specific orientation around the threefold axis). The role of collective JT order in driving inversion symmetry breaking in this system has been identified previously, as has the intriguing possibility of magnetoelectric coupling in the hypothetical chromium(II) analogue $GuaCr(HCOO)_3$[49,50]. Quantum mechanical calculations estimate the maximum polarisation in this system to be $0.22\,\mu C/cm^2$[50], which demonstrates that the achievable polarisations in these molecular perovskites is likely comparable to that of the Rochelle salts, for example[51].

**Propolar molecular perovskites.** Drawing on this concept of inversion symmetry breaking via coupling to collective JT order, we flag the possibility of identifying 'propolar' molecular frameworks: i.e., systems such as $[Gua]M(HCOO)_3$ with pre-existing structural distortions (tilts, multipolar order, …) such that superposition of JT order would be expected to give a polar state. In addition to the guanidinium transition-metal formates discussed above, we identify two additional molecular perovskites that satisfy this criterion for $M_3^+$-type JT order (the most frequently observed). The first is the hypophosphite $[Trz]Mn(H_2PO_2)_3$ (Trz = 1,2,4-triazolium, $C_2N_3H_4^+$)[27], the structure of which is described by a coupling of layered shifts $(X_5^+)$ and conventional $a^-$ tilts $(R_5^-)$. The second is the formate perovskite $[H_2Im]Mn(HCOO)_3$ (HIm = imidazole, $C_3N_2H_4$), which is isostructural and hence described in terms of the same distortion modes[52]. In both cases, substitution of Mn for Cu might reasonably be expected to drive hybrid improper ferroelectricity—a prediction that could be tested experimentally.

A related strategy is to ask, for a given structure type, what particular distortions need to be added to generate bulk polarisation? Whereas the most common space group for conventional perovskites is $Pnma$[53], it is not yet clear what crystal symmetry—if any—is especially prominent amongst molecular perovskites[16]. Hence, we illustrate this strategy with the $Pnma$ perovskite structure, mindful of the possibility of extrapolating to other structures as our collective understanding develops. The key distortions in the $Pnma$ structure are two tilts—one in-phase $(M_2^+)$ and the other out-of-phase $(R_5^-)$. An additional antiferroelectric A-site cation displacement $(X_5^-)$ appears as a secondary order parameter. Consequently, quadrupolar A-site order that transforms as either $R_5^+$ or $X_5^+$ irreps may provide the required coupling with the octahedral rotations $(R_5^-)$ or anti-polar distortions $(X_5^-)$. Alternatively, dipolar ordering transforming as $M_3^-$ or $M_5^-$ could couple to the octahedral rotations at the M-point $(M_2^+)$. Hence, a molecular

**Relevance to polar molecular perovskites.** But to what extent are these predictions borne out in practice? There are in fact a number of hybrid ferroelectric molecular perovskites already reported in the literature. In Table 2 we list the various molecular perovskites known to adopt polar structure types, and present the corresponding irreps responsible for the emergence of polarisation (noting that it is not always possible to identify unambiguously the primary order parameters from a single crystal structure alone). We now proceed to rationalise the emergence of polarisation in detail for two of these systems in light of the analysis presented above. For ease of illustration, we give two-dimensional representations of the relevant coupling schemes in Fig. 3.

Our first experimental example is that of the formate perovskite $[NH_4]Cd(HCOO)_3$, first synthesised several decades ago[42] and revisited more recently for its dielectric properties[43]. The system crystallises in the polar space group $Pna2_1$, with the formate anion adopting a mixed *syn-anti* binding mode. This binding geometry is less prevalent than the *anti-anti* binding mode typically observed for formate-bridged perovskites[20,44–47], and is related to the low tolerance factor[47]. The structure contains a considerable planar shift alternating along the **c** direction which is described by the irrep $X_5^+$. It also supports an unconventional tilt, which transforms as $X_5^-$. These distortions yield the polar space group $Cmc2_1$, which in combination with the conventional in-phase tilting $(M_2^+)$ gives $Pna2_1$. Despite the fact that three order parameters are needed to fully account for the observed space group, the shift $(X_5^+)$ and unconventional tilt $(X_5^-)$ are sufficient to account for inversion symmetry breaking.

Our second example is that of $[Gua]Cu(HCOO)_3$ (where Gua = $C(NH_2)_3^+$), which also crystallises with $Pna2_1$ space group symmetry. In this case the coupling is between a cooperative Jahn–Teller distortion and multipolar A-site order. It is the only polar member of the family $[Gua]M(HCOO)_3$ (M = Mn, Fe, Co, Ni, Cu, Zn, Cd)[44,48], and also the only Jahn–Teller-active member

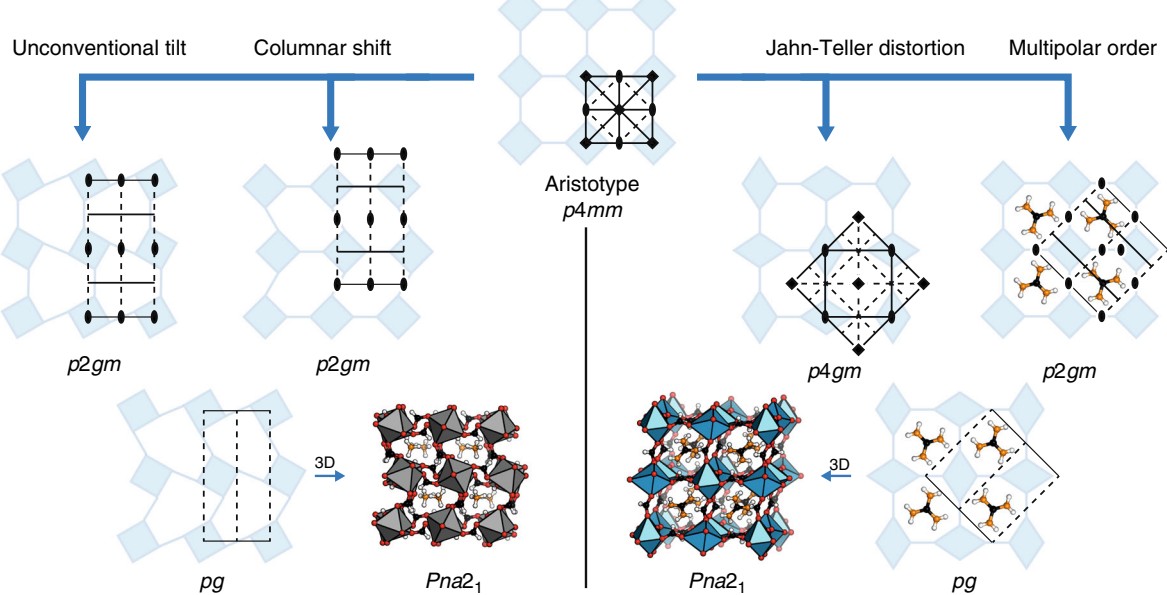

**Fig. 3** Inversion symmetry breaking in 2D molecular perovskites. Left: both unconventional tilting and columnar shifts yield the plane group *p2gm*, but with different origins. When coupled together, the resulting plane group is *pg*, which lacks inversion symmetry. A conceptually related combination of octahedral tilts and shifts is responsible for inversion symmetry breaking in [NH₄]Cd(HCOO)₃, which crystallises in *Pna2₁*. Right: a *C*-type cooperative Jahn–Teller distortion lowers 2D molecular perovskite symmetry to *p4gm*, whereas antiferrohexapolar order gives a *p2gm* cell with a different origin. When combined, the two distortions generate the polar plane group *pg*. This is a 2D analogue of the hybrid coupling found in GuaCu(HCOO)₃

cation with both a dipolar moment and quadrupole moment, enclosed in a *Pnma* perovskite, will give rise to hybrid improper ferroelectricity, provided that the dipole moment orders at the M-point (or, trivially, at the Γ-point) or the quadrupole moment orders at the X- or R-point. To maximise the likelihood of such a situation, one should choose cations with both quadrupolar and dipolar moments, of which there are many examples; e.g., hydrazinium, imidazolium, and methylammonium. Steric considerations will stabilise orientational order of larger cations to lower temperatures, and design rules for controlling the type of quadrupolar order are now starting to emerge[35]. Hence, judicious choice of a molecular cation with the correct type of multipole moment may prove an important design strategy for engineering hybrid improper ferroelectricity.

## Discussion

Hence, we can conclude that, from a group-theoretical viewpoint, perovskites with a molecular component are remarkably predisposed to crystallising in polar space groups. This attractive property is a result of the large number of polar coupling schemes generated by distortions accessible to molecular perovskites but inaccessible to conventional inorganic perovskites. We have identified specific combinations of structural distortions that lead to acentric structures and have thereby suggested possible routes for targeted material design. In particular, we highlight explicitly a number of propolar candidates, where the replacement of JT-inactive for JT-active transition-metal cations is likely to drive polarity. We have also found that either A-site quadrupolar order or the activation of columnar shifts—features that are unique to molecular perovskites—can drive polarisation when coupled to conventional (*Pnma*-type) distortions of the perovskite structure. There are several examples of hybrid improper ferroelectric coupling—although not always recognised as such—already present in the literature and we fully anticipate further examples will continue to emerge. Despite the fact that our analysis has been focussed on polarisation in formate perovskite analogues, the general rules developed here extend to all molecular

perovskites and to all applications where inversion symmetry breaking is necessary. A recent example is the development of non-linear optics based on dicyanamide perovskite chemistry[54].

## Methods

**Group theoretical analysis**. A hypothetical aristotype molecular perovskite was employed as our model system, with the space group $Pm\bar{3}m$ and the (monoatomic) A-site cation at $1a$ $(0, 0, 0)$, B-site cation at $2a$ $\left(\frac{1}{2}, \frac{1}{2}, \frac{1}{2}\right)$, and X-site anion at $6f$ $\left(\frac{1}{2}, \frac{1}{2}, z\right)$. The formula is thus $AB(X_2)_3$. This was used as an input to the web-based software ISODISTORT[55], and the rigid-unit modes and orbital ordering patterns could be identified by inspection.

Multipoles may be classified as transforming as irreps of the rotational group SO(3). In the absence of any external perturbations to the rotational symmetry (i.e., a completely spherically symmetric crystal field), monopoles (angular component of *s* orbital) transform as a singly degenerate irrep, dipoles (*p*) as triply degenerate, quadrupoles (*d*) as 5-fold degenerate, etc. To determine the effect on the irreps of lowering the symmetry of SO(3), such as what occurs at any site symmetry in any crystallographic group, we can use descent of symmetry tables. For quadrupoles, if one or more of the irreps enters the symmetric representation ($A_{1g}$), then we may consider the action of lowering the point group symmetry to have resulted in multipolar order (and this is how we define quadrupolar order in the context of this paper). In the aristotype ABX₃ $Pm\bar{3}m$ structure, quadrupoles (i.e., *d*-orbitals) centred at A/B transform as $E_g$ and $T_{2g}$ while at X they transform as $A_{1g}$, $B_{1g}$, $B_{2g}$, $E_g$. Hence, the problem of determining whether a given irrep of space group $Pm\bar{3}m$ implies quadrupolar order is reduced to determining whether its action as a primary order parameter is sufficient to lower the point-group symmetry at A/B/X such that an additional quadrupole (degree of freedom) enters the totally symmetric representation.

In order to ensure that all relevant irreps for quadrupolar (or multipolar) orders were considered, a dummy atom was placed on $48n$ $(x, y, z)$ in order to mimic electron density in the unit cell at the most general position. We use a setting where the A-site is located at the origin of the perovskite unit cell (see Supplementary Table 1 for conversion to the other setting). For each relevant point of the Brillouin zone, we test in turn the action of an order parameter transforming as one of these irreps, and list its effect on lowering the point group symmetry at A, B, and X, and hence the degree of quadrupolar order to which they correspond. We start by considering the Γ-point irreps, a process which corresponds to ascertaining the relationship between lattice strain and multipolar order. Next, we consider high-symmetry order parameter directions (OPDs) for the zone-boundary irreps which, where possible, we choose such that there are no secondary order parameters. This is important, since many primary order parameters transforming as zone-boundary irreps will imply a lattice strain and hence indirectly induce quadrupolar order at the Γ-point, but do not themselves correspond to such a ordering at the X/M/R-points. Where it is not possible to find a high-symmetry OPD with associated

isotropy subgroup without secondary order parameters (SOPs), a process of elimination is used, subtracting the degree of quadrupolar ordering implied by each SOP in turn. In such a manner it was possible to identify unambiguously which irreps were associated with quadrupolar order.

**Data availability**. All data generated or analysed during this study are included in this published article (and its Supplementary Information files).

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

## Acknowledgements

M.S.S. acknowledges the Royal Commission for the Exhibition of 1851 and the Royal Society for Fellowships. A.L.G. acknowledges the EPSRC (UK) and the ERC for support under grants EP/G004528/2 and 279705, respectively.

## Author contributions

H.L.B.B. and M.S.S. performed the symmetry analysis. M.S.S. and A.L.G. supervised the study. All authors designed the study and contributed to the writing of the paper.

## Additional information

**Competing interests:** The authors declare no competing interests.

