## [Peer Review File · Nature Communications]

Reviewers' comments:

Reviewer #1 (Remarks to the Author):

This manuscript describes a crystallographic study of the possibility of achieving polar characteristics in hybrid organic-inorganic perovskites, based on a group theoretical analysis. The paper is clearly written, the science well motivated, and the conclusions are convincing and concisely summarized. While the use of group theory to analyze the propensity of various crystal structures to yield ferroelectric/polar materials is not in itself novel, its application to the class of hybrid organic-inorganic perovskites (with a molecular substitution instead of a single ion on the A/X sites) is certainly timely, since these materials are currently attracting considerable interest, in particular as potential ferroelectrics, or for non-linear optics, as pointed out by the authors. The findings ("recipes") about the importance of columnar ordering and molecular multipole ordering are especially important, both to rationalize known polar/ferroelectric hybrid structures, and to guide the design of new materials with such properties. Overall, I find that this is a very nice study, worthy of publication in Nature Communications as is.

Reviewer #2 (Remarks to the Author):

The present paper is a theoretical study to understand design strategies to achieve ferroelectricity in hybrid molecular perovskite via a mechanism that has become known as hybrid-improper (note, the word "hybrid" has two different, unrelated meanings wrt to hybrid perovskite vs hybrid-improper. to avoid confusion I will use hybrid only when referring to the mechanism and use only molecular when referring to the type of material).

The authors, given the common types of symmetry-inequivalent distortion modes that are prevalent in molecular perovskites, use group theoretical mechanisms to elucidate the ways in which a hybrid-improper ferroelectric mechanism can be realized. Furthermore, they make a connection from the more abstract/mathematical symmetry questions to the real chemistry of the material system, i.e., a chemically intuitive design strategy that is on solid theoretical grounds. While this question has been worked out in detail for simple inorganic perovskites, to my knowledge this has never been clearly worked out for the case of molecular perovskites. As these types of materials have become of significant interest to the physics and chemistry communities for reasons much broader than just ferroelectricity, such as photovoltaics, understanding chemically intuitive ways to "engineer in" ferroelectricity is an exciting question that would/should be of interest to this large community.

As far as publishing in Nature Communication, I do realize this is completely an editorial decision. I neither strongly recommend, nor do I suggest rejection. I'm sorry I can't be more definitive, but what I can say is that I have no hesitation to recommend publishing in PRL or Advanced Materials.

I have a few points I hope the authors consider:

Figure 1 b, and the discussion of it, is not clear to me. Specifically, having not worked on molecular perovskites, it is not immediately clear to me the difference b/w conventional tilts and unconventional tilts. I realize the connectivity within the molecular perovskite structure allows for greater dofs, but specifically what that motion is, and what allows it to be active in molecular perovskites is not clear to me. For example, is there an analogy to the additional dofs allowed by the reduced connectivity in inorganic Ruddlesden-popper perovskites? Etc..

Page 5, discussion starting with "For inorganic perovskites ..." AS with the previous comment, I think the authors here assume the reader has a much greater familiarity with molecular perovskites than I think most will have. For example, they start off talking about the common distortions in inorganic ABO₃ perovskites that lead to the most common perovskite structure type,

the Pnma structure, which is non-polar. Using this as a starting point, they then suggest how to chemically modify the structure, creating a molecular perovskite, which would be a realization of the hybrid improper ferroelectric mechanism, i.e., creating a ferroelectric from a non-polar starting material. This is a perfecting valid thought process, which helps to make clear the essential physics to realize ferroelectricity, however, what isn't clear to me is if this is a good starting point for the molecular perovskite. What I mean is, is the chemical modification a small perturbation to the system which would allow me to use the simple Pnma as a starting point? One way to understand this is via the question, do most molecular perovskites display the a-a-c+ type rotations as in Pnma? I don't think this is as widely understood as in simple inorganic perovskites.

Page 7, "there are several examples of hybrid improper ferroelectric coupling – although not always recognized as such ..." I think the paper would be much stronger if the authors pointed this compounds out (and references) and at least stated the specific tri-linear coupling in each material that leads to ferroelectric, i.e., one could make a corresponding figure to figure 2 with the known examples. I think this is important for the broader impact of this paper, particularly given that one of the two examples discussed in this paper has already been pointed out to display a hybrid improper mechanism.

Minor:

Page 2, "the mechanism is inherently incompatible with spin-active ..." I think (and I believe most now agree) that the word "incompatible" is too strong of a word. While indeed the filling of the levels involved in a 2nd order Jahn teller mechanism reduces the energy gain from that mechanism, there is no symmetry reason to suggest that the gain is zero and therefore incompatible.

Reviewer #3 (Remarks to the Author):

Recently, hybrid improper ferroelectricity (HIF) attracts widespread interest, because it offers a new route to design novel ferroelectrics, and even near-room-temperature multiferroics (see e.g. Nat. Commun. 5, 4021 (2014)). In inorganic perovskites ABX_3 , there exists one term ($M_{2+} R_{5-} X_{5-}$) coupling the in-phase tilting (M_{2+}), anti-phase tilting (R_{5-}), and A-site antipolar motion (X_{5-}). When designing the perovskites $A'B'O_3/A''B'O_3$ superlattices with A-site layer ordering, the HIF can emerge via the term $(K_{1A'}-K_{1A'}) \omega_R \omega_M P$, which couples the A-site constant term ($K_{1A'}-K_{1A'}$, X_{1+}), the in-phase tilting ω_M (M_{2+}), the anti-phase tilting ω_R (R_{5-}) and the polarization P (Γ_{4-}). The basic idea about HIF in inorganic perovskites have already been shown in the reference (see e.g. Phys. Rev. B 89, 174101 (2014)).

In this work, the authors focus on a large family of "perovskites" (i.e. the molecular hybrid perovskites), and use group theoretical analysis to address a large variety of terms that can give rise to improper ferroelectricity. They consider various possible order parameters, including conventional tilting, unconventional tilting, columnar shift, Jahn-Teller distortion, dipole, quadrupole, and hexapole, etc. They first analyze the irreducible representations of these order parameters. Then, they use the software ISODISTORT to address the possible invariants coupling these order parameters, according to their irreducible representations. I think the main idea is to consider the direct products of different representations, and check if the direct products contain the identical representation (i.e. Γ_1 or Γ_{1+}). At the end of the article, the authors also briefly discussed some reported examples about the hybrid ferroelectric molecular perovskites, including $NH_4Cd(HCOO)_3$ and $GuaCu(HCOO)_3$, using their derived couplings. To summarize, this work is well organized. The idea of this work is quite straightforward and clear. Besides, the various coupling terms in hybrid perovskites may result in a variety of candidates for novel ferroelectrics and multiferroics. The results are interesting, and will provide the guidance to experiments. I also appreciate it that my above mentioned terms $((K_{1A'}-K_{1A'}) \omega_R \omega_M P)$, which is the coupling of $X_{1+} M_{2+} R_{5-} \Gamma_{4-}$, can also be found in the Table S3 of the supplementary materials.

On the other hand, although the authors provide a lot of coupling terms that can lead to HIF in hybrid perovskites, they do not really provide their suggested/predicted candidates that may have hybrid improper ferroelectricity. So, the experimental scientists can not directly verify the theoretical findings. In fact, although there can exist polarization that are allowed by symmetry, this does not really mean that the materials with improper polarization can be easily synthesized experimentally. Let's take the HIF in inorganic perovskites as an example. To obtain the HIF in inorganic perovskites, based on the coupling term $(K1A'-K1A') \omega_R \omega_M P (X1+ M2+ R5- \Gamma4-)$, one should grow the thin film of perovskites $A'BO_3/A''BO_3$ with layering ordering of A' and A'' ions along out-of-plane direction. To obtain the HIF in $A'BO_3/A''BO_3$, the in-phase tilting $\omega_M (M2+)$ should definitely be out-of-plane (i.e. be perpendicular to the substrate rather than be parallel to the substrate). However, in reality, the inorganic perovskites may probably fit with the substrate by the manner that the in-phase tilting is parallel to the substrate (see Phys. Rev. B 94, 024105 (2016)). In this case, the HIF will be quite difficult to obtain. When reading the manuscript, I find that these kinds of things, related to the reality of growing such molecular perovskites with hybrid improper ferroelectricity, are not clear.

About this point, I suggested that the authors should at least do some numerical simulations (e.g. first-principles simulations), to predict the possible candidates that are ferroelectricity based on the couplings in their paper. For example, the authors should show that the configurations that present HIF are indeed favorable energetically than other configurations without HIF. Furthermore, the authors should also tell us what are the values of the candidates' polarization, according to their coupling terms.

Minor comments:

(1) In Fig 1 and Fig S1, the authors show quite a lot of sketches of structural distortions in molecular perovskites. I do not think their sketches are really clear. I suggest that the author can show the polar distortions by a way similar to Figs 1d,e,f of the reference: Nat Commun 8, 14025 (2017).

(2) Although they show various couplings, I do not think their results are really clear to the readers that are not familiar with group theory. So, I suggest that the authors "translate" their couplings terms from the way of combining irreducible representations, to the way of combining order parameters (like what the authors did in the references: Phys. Rev. B 89, 174101 (2014) and Nat Commun 8, 14025 (2017)). Of course, before doing that, the authors may need to give a name of the distortions in their Fig 1 and Fig S1.

(3) If considering my point (2), it is better to define the direction of the order parameters, and show the coupling terms with not only order parameters but also the directions of order parameters. For example, if writing a term as $\sum A_x B_y C_z$, one will know that the appearance of A order parameter along x and B order parameter along y direction will lead to the order parameter C along z direction (rather than y direction or x direction). However, if writing the coupling terms as $\sum ABC$, one will only know that the existence of A and B order parameters will lead to the C order parameter, but without knowing the direction of each order parameter.

The referees raised a number of important points about our manuscript, which we now hope to have addressed in this revised manuscript. A detailed point-by-point response is provided below.

We very much hope our revised manuscript will now be considered suitable for acceptance as an article in Nature Communications.

Reviewer #1 (Remarks to the Author):

This manuscript describes a crystallographic study of the possibility of achieving polar characteristics in hybrid organic-inorganic perovskites, based on a group theoretical analysis. The paper is clearly written, the science well motivated, and the conclusions are convincing and concisely summarized. While the use of group theory to analyze the propensity of various crystal structures to yield ferroelectric/polar materials is not in itself novel, its application to the class of hybrid organic-inorganic perovskites (with a molecular substitution instead of a single ion on the A/X sites) is certainly timely, since these materials are currently attracting considerable interest, in particular as potential ferroelectrics, or for non-linear optics, as pointed out by the authors. The findings (“recipes”) about the importance of columnar ordering and molecular multipole ordering are especially important, both to rationalize known polar/ferroelectric hybrid structures, and to guide the design of new materials with such properties. Overall, I find that this is a very nice study, worthy of publication in Nature Communications as is.

Thank you!

Reviewer #2 (Remarks to the Author):

The present paper is a theoretical study to understand design strategies to achieve ferroelectricity in hybrid molecular perovskite via a mechanism that has become known as hybrid-improper (note, the word “hybrid” has two different, unrelated meanings wrt to hybrid perovskite vs hybrid-improper. to avoid confusion I will use hybrid only when referring to the mechanism and use only molecular when referring to the type of material).

This is a good point, and we have now changed our title to avoid confusion.

The authors, given the common types of symmetry-inequivalent distortion modes that are prevalent in molecular perovskites, use group theoretical mechanisms to elucidate the ways in which a hybrid-improper ferroelectric mechanism can be realized. Furthermore, they make a connection from the more abstract/mathematical symmetry questions to the real chemistry of the material system, i.e., a chemically intuitive design strategy that is on solid theoretical grounds. While this question has been worked out in detail for simple inorganic perovskites, to my knowledge this has never been clearly worked out for the case of molecular perovskites. As these types of materials have become of significant interest to the physics and chemistry communities for reasons much broader than just ferroelectricity, such as photovoltaics, understanding chemically intuitive ways to

“engineer in” ferroelectricity is an exciting questions that would/should be of interest to this large community.

Thank you!

As far as publishing in Nature Communication, I do realize this is completely an editorial decision. I neither strongly recommend, nor do I suggest rejection. I’m sorry I cant be more definitive, but what I can say is that I have no hesitation to recommend publishing in PRL or Advanced Materials.

I have a few points I hope the authors consider:

Figure 1 b, and the discussion of it, is not clear to me. Specifically, having not worked on molecular perovskites, it is not immediately clear to me the difference b/w conventional tilts and unconventional tilts. I realize the connectivity within the molecular perovskite structure allows for greater dofs, but specifically what that motion is, and what allows it to be active in molecular perovskites is not clear to me. For example, is there an analogy to the additional dofs allowed by the reduced connectivity in inorganic Ruddlesden-popper perovskites? Etc..

With regard to unconventional tilts (and as flagged also by referee 3), we have now included an additional panel in Figure 1 and some additional explanatory text in the figure legend. The relationship to RP phases is certainly meaningful and we now cover this briefly in the main text.

Page 5, discussion starting with “For inorganic perovskites ...” AS with the previous comment, I think the authors here assume the reader has a much greater familiarity with molecular perovskites than I think most will have. For example, they start off talking about the common distortions in inorganic ABO₃ perovskites that lead to the most common perovskite structure type, the Pnma structure, which is non-polar. Using this as a starting point, they then suggest how to chemically modify the structure, creating a molecular perovskite, which would be a realization of the hybrid improper ferroelectric mechanism, i.e., creating a ferroelectric from a non-polar starting material. This is a perfecting valid thought process, which helps to make clear the essential physics to realize ferroelectricity, however, what isn’t clear to me is if this is a good starting point for the molecular perovskite. What I mean is, is the chemical modification a small perturbation to the system which would allow me to use the simple Pnma as a starting point? One way to understand this is via the question, do most molecular perovskites display the a-a-c+ type rotations as in Pnma? I don't this is as widely understood as in simple inorganic perovskites.

This is a great point, and it is absolutely the case that it is not yet well understood whether Pnma (or indeed any other specific symmetry) is especially prominent amongst molecular perovskites. Our main point of course was to show that the right sort of unconventional degree of freedom might turn a commonly-accessible centrosymmetric structure into a polar structure. So we have now spent some time elaborating on this point. We do this in three ways: we cover two explicit examples of molecular perovskites where specific experimentally-realizable centrosymmetric structures would become polar with collective Jahn Teller order of the right type (we call them ‘pro-polar’ structures), and we also take the conventional perovskite Pnma structure type and summarise what sort of order of unconventional degrees of freedom would drive polarisation in this structure type.

Page 7, “there are several examples of hybrid improper ferroelectric coupling – although not always recognized as such ...” I think the paper would be much stronger if the authors pointed this compounds out (and references) and at least stated the specific tri-linear coupling in each material that leads to ferroelectric, i.e., one could make a corresponding figure to figure 2 with the known examples. I think this is important for the broader impact of this paper, particularly given that one of the two examples discussed in this paper has already been pointed out to display a hybrid improper mechanism.

This is a great point, and we have now included a summary of the known polar molecular perovskites (Table 2) together with some discussion of the inversion-symmetry-breaking mechanism in specific cases.

Minor:

Page 2, "the mechanism is inherently incompatible with spin-active ..." I think (and I believe most now agree) that the word "incompatible" is too strong of a word. While indeed the filling of the levels involved in a 2nd order Jahn teller mechanism reduces the energy gain from that mechanism, there is no symmetry reason to suggest that the gain is zero and therefore incompatible.

We are sorry for this overstatement and have corrected this in the revised manuscript.

Reviewer #3 (Remarks to the Author):

Recently, hybrid improper ferroelectricity (HIF) attracts widespread interest, because it offers a new route to design novel ferroelectrics, and even near-room-temperature multiferroics (see e.g. Nat. Commun. 5, 4021 (2014)). In inorganic perovskites ABX_3 , there exists one term ($M_{2+} R_{5-} X_{5-}$) coupling the in-phase tilting (M_{2+}), anti-phase tilting (R_{5-}), and A-site antipolar motion (X_{5-}). When designing the perovskites $A'BO_3/A''BO_3$ superlattices with A-site layer ordering, the HIF can emerge via the term $(K_{1A'}-K_{1A''}) \omega_R \omega_M P$, which couples the A-site constant term ($K_{1A'}-K_{1A''}$, X_{1+}), the in-phase tilting ω_M (M_{2+}), the anti-phase tilting ω_R (R_{5-}) and the polarization P (Γ_{4-}). The basic idea about HIF in inorganic perovskites have already been shown in the reference (see e.g. Phys. Rev. B 89, 174101 (2014)).

In this work, the authors focus on a large family of "perovskites" (i.e. the molecular hybrid perovskites), and use group theoretical analysis to address a large variety of terms that can give rise to improper ferroelectricity. They consider various possible order parameters, including conventional tilting, unconventional tilting, columnar shift, Jahn-Teller distortion, dipole, quadrupole, and hexapole, etc. They first analyze the irreducible representations of these order parameters. Then, they use the software ISODISTORT to address the possible invariants coupling these order parameters, according to their irreducible representations. I think the main idea is to consider the direct products of different representations, and check if the direct products contain the identical representation (i.e. Γ_1 or Γ_{1+}). At the end of the article, the authors also briefly discussed some reported examples about the hybrid ferroelectric molecular perovskites, including $NH_4Cd(HCOO)_3$ and $GuaCu(HCOO)_3$, using their derived couplings. To summarize, this work is well organized. The idea of this work is quite straightforward and clear. Besides, the various coupling terms in hybrid perovskites may result in a variety of candidates for novel ferroelectrics and multiferroics. The results are interesting, and will provide the guidance to experiments. I also appreciate it that my above mentioned terms $((K_{1A'}-K_{1A''}) \omega_R \omega_M P)$, which is the coupling of $X_{1+} M_{2+} R_{5-} \Gamma_{4-}$, can also be found in the Table S3 of the supplementary materials.

Thank you!

On the other hand, although the authors provide a lot of coupling terms that can lead to HIF in hybrid perovskites, they do not really provide their suggested/predicted candidates that may have hybrid improper ferroelectricity. So, the experimental scientists can not directly verify the theoretical findings. In fact, although there can exist polarization that are allowed by symmetry, this does not really mean that the materials with improper polarization can be easily synthesized experimentally. Let's take the HIF in inorganic perovskites as an example. To obtain the HIF in inorganic perovskites, based on the coupling term $(K_{1A'}-K_{1A''}) \omega_R \omega_M P$ ($X_{1+} M_{2+} R_{5-} \Gamma_{4-}$), one should grow the thin film of perovskites $A'BO_3/A''BO_3$ with layering ordering of A' and A'' ions along out-of-plane direction. To obtain the HIF in $A'BO_3/A''BO_3$, the in-phase tilting ω_M (M_{2+}) should definitely be out-of-plane (i.e. be perpendicular to the substrate rather than be parallel to the substrate). However, in reality, the inorganic perovskites may probably fit with the substrate by the manner that

the in-phase tilting is parallel to the substrate (see Phys. Rev. B 94, 024105 (2016)). In this case, the HIF will be quite difficult to obtain. When reading the manuscript, I find that these kinds of things, related to the reality of growing such molecular perovskites with hybrid improper ferroelectricity, are not clear.

We are sorry for not having made clearer the known experimental realisations of the HIF mechanism we discuss. We have now included Table 2, which summarises all known examples, including the corresponding HIF couplings, and we discuss in detail a number of these in the main text. In addition we also make some clear predictions for new materials that should exhibit HIF on the basis of the theory we set out.

About this point, I suggested that the authors should at least do some numerical simulations (e.g. first-principles simulations), to predict the possible candidates that are ferroelectricity based on the couplings in their paper. For example, the authors should show that the configurations that present HIF are indeed favorable energetically than other configurations without HIF. Furthermore, the authors should also tell us what are the values of the candidates' polarization, according to their coupling terms.

The referee makes an excellent point, and clearly further computational studies will be of significant interest and value in the field. Our present study is conceptual and instead of developing our own calculations we draw instead on the existing experimental and computational work of others. This includes calculations of the polarisation accessible in some of these systems, which we now include explicitly and compare with that achievable in other ferroelectric materials.

Minor comments:

(1) In Fig 1 and Fig S1, the authors show quite a lot of sketches of structural distortions in molecular perovskites. I do not think their sketches are really clear. I suggest that the author can show the polar distortions by a way similar to Figs 1d,e,f of the reference: Nat Commun 8, 14025 (2017).

We have now included a new panel to Figure 1, which we have constructed along the lines of the Figure referenced by the referee. We hope that this representation, including the accompanying explanatory text, will do a better job of bridging our study to those of ferroelectric mechanisms in conventional perovskites.

(2) Although they show various couplings, I do not think their results are really clear to the readers that are not familiar with group theory. So, I suggest that the authors "translate" their couplings terms from the way of combining irreducible representations, to the way of combining order parameters (like what the authors did in the references: Phys. Rev. B 89, 174101 (2014) and Nat Commun 8, 14025 (2017)). Of course, before doing that, the authors may need to give a name of the distortions in their Fig 1 and Fig S1.

We hope that the new panel Fig1(c), together with a clearer description of the coupling mechanism in known materials (couched in terms of tilts, or shifts, or multipolar order) will help address this point.

(3) If considering my point (2), it is better to define the direction of the order parameters, and show the coupling terms with not only order parameters but also the directions of order parameters. For example, if writing a term as $\sum A_x B_y C_z$, one will know that the appearance of A order parameter along x and B order parameter along y direction will lead to the order parameter C along z direction (rather than y direction or x direction). However, if writing the coupling terms as $\sum ABC$, one will only know that the existence of A and B order parameters will lead to the C order parameter, but without knowing the direction of each order parameter.

The referee makes a very important point here. We absolutely agree that a full understanding of these couplings from the group theoretical viewpoint requires consideration both of the irreducible

representations and the corresponding order parameter directions. Indeed we hope to follow this brief communication with a comprehensive in-depth analysis of precisely this point and the implications for specific order parameter directions for specific coupling schemes. Given the scope for ready descent into complex group theoretical arguments, the limitations on space in a Nature Communications article, and the need to remain mindful of a general scientific audience, we hope that the referee will understand our preference to retain discussion of these details for a specialised in-depth follow-up article.

REVIEWERS' COMMENTS:

Reviewer #2 (Remarks to the Author):

I have read the revised manuscript, and the author's responses to Referees 1-3. I find the authors have nicely addressed the concerns raised. I reiterate a point from my original report in that this paper addresses "... an exciting questions that would/should be of interest to this large community." Furthermore, I am now convinced Nature Comm is the appropriate journal for publication.

Reviewer #3 (Remarks to the Author):

The authors have only partially addressed my comments. For example, they have not addressed my minor point (3). Besides, I believe that the numerical simulations (i.e. first-principles simulations) will be quite helpful for the experimental researchers to obtain novel HIF materials, based on their group theoretical results. However, the authors have not used numerical simulations to predict new candidates, based on their coupling terms. On the other hand, as a conceptual study, the results are quite interesting and important. I have already mentioned this point in my previous report.

Whether to recommend this work (for publication) or not, it is a question for me. Clearly, the editor's decision will help me solve this problem.